# SEMI-SUPERVISED FEW-SHOT LEARNING WITH MAML

**Rinu Boney & Alexander Ilin**
The Curious AI Company
{rinu,alexilin}@cai.fi

## ABSTRACT

We present preliminary results on extending Model-Agnostic Meta-Learning (MAML) (Finn et al., 2017a) to fast adaptation to new classification tasks in the presence of unlabeled data. Using synthetic data, we show that MAML can adapt to new tasks without any labeled examples (unsupervised adaptation) when the new task has the same output space (classes) as the training tasks do. We further extend MAML to the semi-supervised few-shot learning scenario, when the output space of the new tasks can be different from the training tasks.

## 1 INTRODUCTION

We consider the problem of fast adaptation to new classification tasks in the presence of unlabeled data. This problem is topical in many practical applications: As an example, consider a material recognition system deployed in multiple factories. The recognition tasks at different sites often share the output space, that is the same categories of materials need to be recognized. However, factories may have slightly different lighting conditions or other factors affecting the recognition process, so there is a need for fast adaptation of every recognition system. Oftentimes, the output space also differs across factories but it is still desirable to transfer the knowledge from old recognition tasks to the new ones. In many such situations, collecting unlabeled data is cheap while labeling the data is laborious, therefore it is highly desirable to be able to adapt from a few labeled examples. We call this problem semi-supervised few-shot learning and here we show how it can be addressed by the recently introduced Model-Agnostic Meta-Learning (MAML, Finn et al., 2017a).

## 2 MAML ADAPTATION WITH THE USE OF UNLABELED DATA

Suppose there exists a set of related classification tasks in which each $i$-th task is described by data $D^{(i)} = \{(\mathbf{x}_j, y_j)\}$ with inputs $\mathbf{x}_j$ and targets $y_j$. In MAML, a common classifier $y \approx f(\mathbf{x}, \boldsymbol{\theta})$ is adapted to task $i$ by updating its parameters $\boldsymbol{\theta}$ using one or more gradient descent steps:

$$\boldsymbol{\theta}_i = \boldsymbol{\theta}_* - \boldsymbol{\alpha} \sum_{(\mathbf{x}_j, y_j) \in D_{\text{train}}^{(i)}} \nabla_{\boldsymbol{\theta}} L\left(f(\mathbf{x}_j, \boldsymbol{\theta}), y_j\right)\bigg|_{\boldsymbol{\theta} = \boldsymbol{\theta}_*}, \tag{1}$$

where $L$ is the loss function computed on the training samples $D_{\text{train}}^{(i)}$. The initial values $\boldsymbol{\theta}_*$ and the vector of learning rates $\boldsymbol{\alpha}$ are the parameters tuned during MAML training. Training happens by going through a set of tasks, adapting the classifier to each task using (1) and updating the parameters $\boldsymbol{\theta}_*, \boldsymbol{\alpha}$ so as to optimize the performance of the adapted models $f(\mathbf{x}, \boldsymbol{\theta}_i)$ on the validation sets $D_{\text{val}}^{(i)}$. This optimization is performed using standard backpropagation, which involves a gradient through a gradient since the computation of $\boldsymbol{\theta}_i$ contains $\nabla_{\boldsymbol{\theta}} L$.

**Unsupervised adaptation.** We first consider the case when the output space of the tasks stays constant but the input distributions of tasks vary. In this case, the adaptation to new tasks may be done in a purely unsupervised manner without the use of labeled examples from the new task. We perform classification in two steps: 1) inputs $\mathbf{x}$ are transformed into features $\mathbf{z}$ using function $\mathbf{z} = f(\mathbf{x}, \boldsymbol{\theta})$ with parameters $\boldsymbol{\theta}$; 2) predicted labels are computed using function $\hat{y} = g(\mathbf{z}, \boldsymbol{\omega})$ with parameters $\boldsymbol{\omega}$. The idea is that when a new task contains only unlabeled data, one can use MAML to

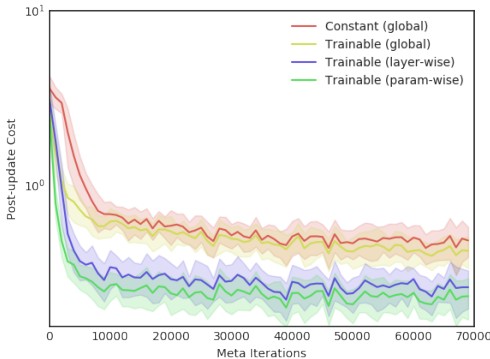

Figure 1: Learning curves of MAML on the toy sine regression dataset from (Finn et al., 2017a) for different learning rate couplings. The constant learning rate was reported by the original paper.

adapt only the feature extractor $f$ and keeping the classifier $g$ constant. The adaptation can be done with a gradient-based based rule similar to (1):

$$\boldsymbol{\theta}_i = \boldsymbol{\theta}_* - \boldsymbol{\alpha} \sum_{\mathbf{x}_j \in D_{\text{train}}^{(i)}} \nabla_{\boldsymbol{\theta}} C\left(f(\mathbf{x}_j, \boldsymbol{\theta})\right)\bigg|_{\boldsymbol{\theta}=\boldsymbol{\theta}_*}, \tag{2}$$

where $C$ is some kind of a cost function that measures the quality of the extracted features. Several auxiliary cost functions using unlabeled data have been proposed in the literature to improve the classification performance in the semi-supervised scenario (see, e.g., Grandvalet & Bengio, 2005; Rasmus et al., 2015; Miyato et al., 2015; Laine & Aila, 2016; Tarvainen & Valpola, 2017). Instead of specifying that extra cost, we propose to parametrize it with a neural network $C(\mathbf{z}, \boldsymbol{\phi})$ with parameters $\boldsymbol{\phi}$ and learn it in the same meta-training procedure. Similar to the supervised case, the parameters are tuned to optimize the performance on the validation sets:

$$\min_{\boldsymbol{\theta}_*, \boldsymbol{\alpha}, \boldsymbol{\omega}, \boldsymbol{\phi}} \sum_i \sum_{(\mathbf{x}_j, y_j) \in D_{\text{val}}^{(i)}} L\left(g(f(\mathbf{x}_j, \boldsymbol{\theta}_i), \boldsymbol{\omega}), y_j\right). \tag{3}$$

Note that this learning algorithm is essentially equivalent to the two-head architecture proposed by Finn et al. (2017b) for imitation learning.

**Semi-supervised adaptation** is relevant for problems when the output space varies across different tasks. Since the output space of the classification task can change, we now need to adapt both the feature extractor $f$ and the classifier $g$. We again first use unlabeled data to adapt the feature extractor using (2) with a meta-trained cost function $C$. And finally we use labeled data to adapt the classifier parameters $\boldsymbol{\omega}$ using an update rule similar to (1). The difference to the fully supervised case is that the loss function is computed using the adapted features $\mathbf{z}$ instead of raw inputs $\mathbf{x}$. Again, the parameters are tuned to optimize the performance on the validation sets $D_{\text{val}}^{(i)}$. The tuned parameters are $\boldsymbol{\theta}_*, \boldsymbol{\alpha}, \boldsymbol{\phi}, \boldsymbol{\omega}$ and $\boldsymbol{\alpha}_{\boldsymbol{\omega}}$ which is the learning rate for adapting $\boldsymbol{\omega}$.

**Importance of decoupling learning rates.** Note that using different learning rates $\boldsymbol{\alpha}$ and $\boldsymbol{\alpha}_{\boldsymbol{\omega}}$ was crucial to make this approach work. This could be attributed to the difficulty in balancing the training signals provided by the labeled and unlabeled samples. In the method proposed here, the unsupervised cost is forced to be developed because the feature extraction part of the network is solely adapted using the unsupervised loss. We explored different learning rate schemes in MAML and found layer-wise or parameter-wise learning rates to be an important factor in balancing the signals. Fig. 1 illustrates that decoupling learning rates can significantly improve the performance of adapted models, which supports the results reported by Li et al. (2017). Useful insights on the role of the learning rates in MAML were recently provided by Anonymous (2018).

## 3 EXPERIMENTS WITH SYNTHETIC DATA

We tested the proposed algorithms on a synthetic dataset from (Boney & Ilin, 2017). The dataset consists of a set of two-dimensional classification tasks with two classes, in which the optimal

decision boundary is a sine wave (see Fig. 2). The amplitude $A$ of the optimal decision boundary varies across tasks within $[0.1, 5.0]$ and the phase $\psi$ varies within $[0, 2\pi]$. The first dimension of the data samples is drawn uniformly from $[-5, 5]$ and the second dimension is computed as $x_2 = A\sin(x_1 + \psi) + c$, where $c$ is a noise term with the Laplace distribution with the mean $\pm 2$ (depending on the class) and the scale parameter 0.5. We sampled 100 tasks for training and 1000 for testing.

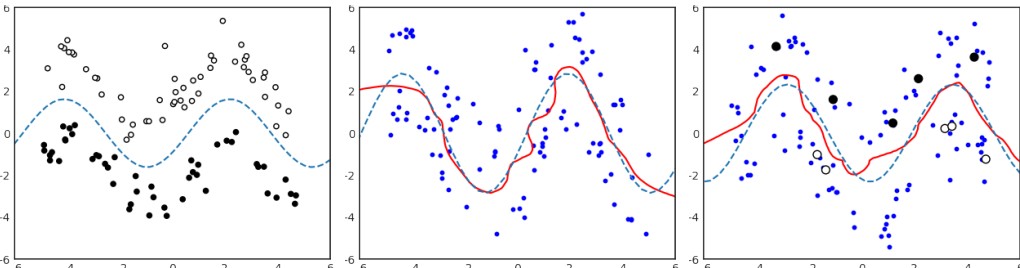

Figure 2: Examples of adaptation in the sine dataset. The black dots correspond to samples of one class and the white dots correspond to samples from the other class. The blue dots correspond to unlabeled samples. The optimal decision boundary is shown with the dashed blue line. The red line depicts the decision boundaries of the adapted model. Left: Example task. Middle: Example of unsupervised adaptation to a test task from 100 unlabeled samples. Right: Example of semi-supervised adaptation to a test task from 10 labeled samples and 100 unlabeled samples.

Examples of the decision boundaries produced by MAML on test tasks are shown in Fig. 2. Note that even for a small number of labeled examples the adapted decision boundaries resemble the sine wave, thus the knowledge is transferred between tasks. The classification accuracy of the models adapted with MAML are shown in Fig. 3. We compare the MAML approach to semi-supervised few-shot learning with Prototypical Networks (PN), as proposed in (Boney & Ilin, 2017). We use a fully connected network with two hidden layers of size 100 with ReLU nonlinearity as the feature extractor and the sum of squares of a two dimensional linear projection as the parametrization of the unsupervised cost. One can see that in this experiment PN generally performs better in the fully supervised setting, while MAML is much more efficient in making use of unlabeled data. In fact, its performance was very close to the fully supervised case using true labels of the unlabeled samples.

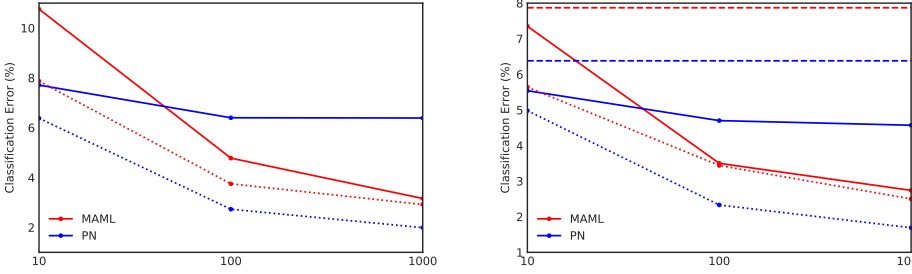

Figure 3: Classification errors of adapted models as functions of the number $n$ of unlabeled samples. Left: Unsupervised adaptation. The dotted lines depicts the accuracy of the adapted model using the same number $n$ of *labeled* samples. Right: Semi-supervised adaptation with 10 labeled plus $n$ unlabeled samples. The dashed and dotted lines depicts the error rates of the (fully supervised) adapted model using 10 and $n + 10$ *labeled* samples respectively.

## 4    DISCUSSION AND FUTURE WORK

In this work, we proposed an extension of MAML to the cases of unsupervised and semi-supervised few-shot adaptation. Using a synthetic dataset, we show that MAML can be more efficient in using unlabeled data compared to other techniques. We continue investigating whether the proposed approach is practical for larger datasets and how to combine the good properties of MAML and PN in a single adaptation scheme.

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
