# OpenReview forum: "Semi-Supervised Few-Shot Learning with MAML"
_ICLR.cc/2018/Workshop — Accept_

### Official Review · AnonReviewer4 · 2018-03-07
**A method for fast adaptation with unlabeled data. The method uses gradient-based procedures with different learning rates for supervised and unsupervised informations. It is hard for me to really capture the novelty of this approach. The use of 2 learning rates and also the capability to deal with few shot learning problems claimed by the authors are not sufficiently discussed and evaluated.**

**Rating:** 5
**Confidence:** 2

**Review:**

Summary:
--------
The paper proposes a preliminary method for fast adaptation to new classification tasks in the presence of unlabeled data.
The idea is to optimize a parameterized function - such as neural network - that generates new features and to control the features generated by an appropriate cost function. The optimization is done by gradient-based rule from the initial parameter values.
When supervised information are available, a classifier can be jointly learned with the previous formulation with a different learning rate.
An experiments on synthetic data.

Comment:
The presentation of the contribution is unclear to me and it is not obvious to see the novelty with respect to state of the art methods that already tried to learn better representation of the data.
The authors claim that their approach can be interesting for few shot learning scenarios, but I am unable to understand where the paper bring a novelty here.
The important point seems the use of different learning rates for the unsupervised and supervised part - for me this is not really new - adversarial based architectures already use different learning rates.
I think that a better presentation of the contribution with respect to state of the art would be useful. A better study/discussion on the different learning rates seems necessary.


Pro
-method that aims to adapt quickly a model taking into account unsupervised data using gradient based procedure with different learning rates.
Cons
-It is difficult for me to see the novelty of the contribution in the way the paper is written
-The aspects related to few shot learning and use of different learning rates are not sufficiently discussed and evaluated.

---

### Official Review · AnonReviewer1 · 2018-03-08

**Rating:** 7
**Confidence:** 5

**Review:**

This paper considers the setting of learning to learn from unsupervised or semi-supervised data. It builds upon the MAML algorithm to do so, and additionally provides some useful empirical insights into the effect of additionally learning the learning rate.

Pros:
- learning to learn from unsupervised or semi-supervised data is an important and relatively unexplored topic
- the paper presents a systematic comparison between MAML and prototypical networks on a toy problem in this setting.
- useful insights on the effect of adapting the learning rate with the MAML algorithm

Cons:
- simple experimental domain (but I would expect the conclusions to transfer to more complex domains given that the MAML and PN algorithms have both been demonstrated in complex domains, such as image recognition)

I have a few questions and suggestions for improvements:
1. Does the layer-wise learning rates treat the weight and bias of a given layer as different layers or the same? If it treats them as the same, does it help to split them up?
Also, it would helpful to list the number of total parameters for each of the lines in Figure 1. While this plot nicely shows the affect of this decision on the training error, there may be different effect in settings with limited training data and where overfitting is a concern.
2. In figure 1, (a) How many gradient steps were used?, and (b) Was a bias transformation used? (as proposed by Finn et al. 2017b). How does (a) and (b) affect the learning rate effect? [e.g. adapting learning rates leads to additional expressivity, as do (a) and (b). It would be interesting to see if you have a similar effect, or complementary effect of using (a) and (b) vs. using (a) and (b) with learning rate adaptation)
3. In the final version of the paper, it would make sense to cite [1] and [2], as both are quite relevant. [I do not penalize the authors for not citing these works earlier, given that both are quite recent.]

[1] https://openreview.net/forum?id=HJcSzz-CZ
[2] https://arxiv.org/abs/1802.01557

---

### Official Review · AnonReviewer2 · 2018-03-11
**Paper is well-written while there lacks some insightful analysis about the proposed approach and the empirical studies.**

**Rating:** 6
**Confidence:** 4

**Review:**

The authors studied the problem of MAML in presence of unlabeled data/ few-shot labeled data. The paper is well-written and well-organized. The proposed method is straightforward,  while there lacks some insightful analysis about the proposed approach and the empirical studies. In particular,
(1) The authors should clearly identify their contributions beyond the base algorithm (MAML). It would help the readers with less background to appreciate the work.
(2) The authors may want to provide more discussion in depth regarding why the proposed methods perform better than the baseline method (PN) on the synthetic dataset.

---

### Decision · Program_Chairs · 2018-03-20
**ICLR 2018 Workshop Acceptance Decision**

**Decision:**

Accept

**Comment:**

Congratulations, your paper was accepted to the ICLR workshop.